# Impact of Chronic Bronchial Infection by *Staphylococcus aureus* on Bronchiectasis

**DOI:** 10.3390/jcm11143960

**Published:** 2022-07-07

**Authors:** Marta García Clemente, Casilda Olveira, Rosa Girón, Luis Máiz, Oriol Sibila, Rafael Golpe, Rosario Menéndez, Juan Rodríguez, Esther Barreiro, Juan Luis Rodríguez Hermosa, Concepción Prados, David De la Rosa, Claudia Madrid Carbajal, Marta Solís, Miguel Ángel Martínez-García

**Affiliations:** 1Respiratory Department, Central University Hospital, 33011 Oviedo, Spain; claudiamadrid9@hotmail.com; 2Instituto de Investigación Biosanitaria del Principado de Asturias (ISPA), 33011 Oviedo, Spain; 3Respiratory Department, Hospital Regional Universitario de Málaga, 29009 Málaga, Spain; casi1547@separ.es; 4Instituto de Investigación Biomédica de Málaga (IBIMA), Universidad de Málaga, 29003 Málaga, Spain; 5Respiratory Department, Instituto de Investigación Sanitaria, Hospital Universitario de la Princesa, 28015 Madrid, Spain; mgiron@gmail.com (R.G.); msolis1995@gmail.com (M.S.); 6Respiratory Department, Hospital Ramón y Cajal, 28015 Madrid, Spain; luis.maiz@salud.madrid.org; 7Respiratory Department, Hospital Clinic, 08035 Barcelona, Spain; osibila@clinic.cat; 8Centro de Investigación en Red de Enfermedades Respiratorias (CIBERES), Instituto de Salud Carlos III (ISCIII), 28015 Madrid, Spain; mianmartinezgarcia@gmail.com; 9Respiratory Department, Hospital Lucus Augusti, 27080 Lugo, Spain; rafa898@separ.es; 10Respiratory Department, Hospital Universitario y Politécnico La Fe, 46003 Valencia, Spain; rosmenend@gmail.com; 11Respiratory Department, Hospital San Agustín, 33401 Avilés, Spain; juan_rodriguezl@hotmail.com; 12Respiratory Department, Hospital del Mar-IMIM, Parc de Salut Mar, 08003 Barcelona, Spain; ebarreiro@imim.es; 13Pulmonary Department, Research Institute of Hospital Clínico San Carlos (IdISSC), 28040 Madrid, Spain; jlrhemosa@yahoo.es; 14Respiratory Department, Hospital Clinico San Carlos, 28015 Madrid, Spain; 15Department of Medicine, Universidad Complutense de Madrid, 28015 Madrid, Spain; 16Respiratory Department, Hospital La Paz, 28046 Madrid, Spain; conchaprados@gmail.com; 17Respiratory Department, Hospital Santa Creu I Sant Pau, 08041 Barcelona, Spain; david.rosa23@gmail.com

**Keywords:** bronchiectasis, chronic bronchial infection, *Staphylococcus aureus*

## Abstract

The objective of the study was to analyze the factors associated with chronic bronchial infection (CBI) due to methicillin-susceptible *Staphylococcus aureus* (SA) and assess the clinical impact on severity, exacerbations, hospitalizations, and loss of lung function compared to patients with no isolation of PPMs in a large longitudinal series of patients from the Spanish bronchiectasis registry (RIBRON). Material and methods: A prospective, longitudinal, multicenter study was conducted with patients included in the RIBRON registry between January 2015 and October 2020. The inclusion criteria were an age of 18 years or older and an initial diagnosis of bronchiectasis. Patients recorded in the registry had a situation of clinical stability in the absence of an exacerbation in the four weeks before their inclusion. All patients were encouraged to provide a sputum sample at each visit for microbiological culture. Annual pulmonary function tests were performed according to the national spirometry guidelines. Results: A total of 426 patients were ultimately included in the study: 77 patients (18%) with CBI due to SA and 349 (82%) who did not present any isolation of PPMs in sputum. The mean age was 66.9 years (16.2), and patients 297 (69.7%) were female, with an average BMI of 25.1 (4.7) kg/m^2^ and an average Charlson index of 1.74 (1.33). The mean baseline value of FEV1 2 L was 0.76, with a mean FEV1% of 78.8% (23.1). One hundred and seventy-two patients (40.4%) had airflow obstruction with FEV1/FVC < 0.7. The mean predictive FACED score was 1.62 (1.41), with a mean value of 2.62 (2.07) for the EFACED score and 7.3 (4.5) for the BSI score. Patients with CBI caused by SA were younger (*p* < 0.0001), and they had a lower BMI (*p* = 0.024) and more exacerbations in the previous year (*p* = 0.019), as well as in the first, second, and third years of follow-up (*p* = 0.020, *p* = 0.001, and *p* = 0.018, respectively). As regards lung function, patients with CBI due to SA had lower levels of FEV1% at the time of inclusion in the registry (*p* = 0.021), and they presented more frequently with bronchial obstruction (*p* = 0.042). A lower age (OR: 0.97; 95% CI: 0.94–0.99; *p* < 0.001), lower FEV1 value% (OR: 0.98; 95% CI: 0.97–0.99; *p* = 0.035), higher number of affected lobes (OR: 1.53; 95% CI: 1.2–1.95; *p* < 0.001), and the presence of two or more exacerbations in the previous year (OR: 2.33; 95% CI: 1.15–4.69; *p* = 0.018) were observed as independent factors associated with CBI due to SA. The reduction in FEv1% in all patients included in the study was −0.31%/year (95% CI: −0.7; −0.07) (*p* = 0.110). When the reduction in FEv1% is analyzed in the group of patients with CBI due to SA and the group without pathogens, we observed that the reduction in FEV1% was −1.19% (95% CI: −2.09, −0.69) (*p* < 0.001) in the first group and −0.02% (95% CI: −0.07, −0.01) (*p* = 0.918) in the second group. According to a linear regression model (mixed effects) applied to determine which factors were associated with a more pronounced reduction in FEv1% in the overall group (including those with CBI due to SA and those with no PPM isolation), age (*p* = 0.0019), use of inhaled corticosteroids (*p* = 0.004), presence of CBI due to SA (*p* = 0.007), female gender (*p* < 0.001), and the initial value of FEV1 (*p* < 0.001) were significantly related. Conclusions: Patients with non-CF bronchiectasis with CBI due to SA were younger, with lower FEV1% values, more significant extension of bronchiectasis, and a higher number of exacerbations of mild to moderate symptoms than those with no PPM isolation in respiratory secretions. The reduction in FEV1% was −1.19% (95% CI: −2.09, −0.69) (*p* < 0.001) in patients with CBI caused by SA.

## 1. Introduction

Bronchiectasis is an irreversible dilation of the airways that gives rise to characteristic symptoms such as coughing, profuse production of thick mucus, and recurrent bronchial infections. As many patients tend to present with chronic bronchial infection (CBI) and frequent exacerbations, they frequently find themselves involved in a vicious cycle of local (usually neutrophilic but also eosinophilic) and systemic inflammatory infection, as well as progressive airway damage [1,2,3,4,5,6,7,8]. It is well known that patients with bronchiectasis who also present with CBI due to *Pseudomonas aeruginosa* (PA) have more severe disease and a worse prognosis [9,10,11,12]. However, less data are available on other potentially pathogenic microorganisms (PPMs) that can chronically infect the airways of these patients. *Staphylococcus aureus* (SA), a Gram-positive, rod-shaped bacterium, is isolated in bronchiectasis with relative frequency [13,14]. A significant amount of information on the pathogenic and predictive power of SA has been obtained from studies carried out in patients with cystic fibrosis (CF), in whom these isolations are widespread throughout the natural history of the disease [15]. However, there is no information in the scientific literature from large studies on the characteristics and impact of the presence of CBI caused by SA in patients with non-CF bronchiectasis. Therefore, the objective of our study was to analyze the factors associated with CBI due to methicillin-susceptible SA and assess the clinical impact of this microbiological situation on severity, exacerbations, hospitalizations, and loss of lung function compared to patients with no isolation of PPMs in a large longitudinal series of patients from the Spanish bronchiectasis registry (RIBRON).

## 2. Materials and Methods

### 2.1. Design

A prospective, longitudinal, multicenter study was conducted. Forty-three Spanish centers participated in the study, all of which had patients included in the RIBRON registry between January 2015 and October 2020 [16]. An external company (CRO) continuously monitored the quality of the data entered. The study followed the Strengthening the Reporting of Observational Studies in Epidemiology (STROBE) guidelines [17]. All patients were informed of the purpose of the registry, and they gave their written informed consent at their corresponding participating center, as provided by the local ethics committee affiliated with the registry (number: 001-2012, Hospital Josep Trueta, Girona, Spain). The research study was carried out according to the guidelines of the World Medical Association for Research in Humans (7th revision of the Declaration of Helsinki, Fortaleza, Brazil 2013) [18].

### 2.2. Population

Figure 1 shows a flow chart of patient recruitment for the study. The inclusion criteria for the RIBRON registry were an age of 18 years or older and an initial diagnosis of bronchiectasis according to the recently published consensus (diagnosis by high-resolution CT scan and clinical evidence compatible with bronchiectasis) [2,16,19,20,21]. Patients recorded in the registry had a situation of clinical stability in the absence of an exacerbation in the four weeks prior to their inclusion in the registry, and no patients had other diseases related to the pathogen.

All patients were encouraged to provide a sputum sample at each visit for microbiological culture. Following the same rules as other studies published with data from RIBRON, we defined CBI as the presence of three or more consecutive positive cultures for the same PPM, always excluding an exacerbation period [2,22]. We did not take into account isolated cultures of SA that we considered colonization. For the purpose of this study, the performance of at least three spirometry tests with no less than a one-year separation was also a criterion for inclusion. Patients with less than two sputum samples for microbiological culture per year were excluded.

Patients with traction bronchiectasis, patients with CF, patients with CBI due to pathogens other than SA, patients who presented a single isolation of SA and therefore did not meet the criteria for CBI, patients who died in the first year of inclusion in the RIBRON registry, and patients who had only the baseline visit recorded in the registry were excluded from the study.

A total of 426 patients were ultimately included in the study, 77 of whom presented with CBI caused by SA (18%), leaving 349 patients who did not show any isolation of PPMs in sputum samples (82%) during follow-up.

### 2.3. Variables

The following variables and parameters were collected: anthropometric data (age, gender, and body mass index [BMI]), etiology, clinical data (including number and severity of exacerbations), smoking history, lung function, bacterial CBI data, radiological data, number of exacerbations and hospitalizations in the previous year and the three years following inclusion in the registry, comorbidities (Charlson index), and severity score data (FACED, EFACED and BSI) [23,24,25].

Annual pulmonary function tests were performed according to the national spirometry guidelines [26], provided patients were in a situation of clinical stability. Three measurements were made, taking the highest value after administering salbutamol (400 µg). The spirometer was regularly calibrated according to the manufacturer’s recommendations. An external company reviewed all the measurements every month, contacting the researcher if data were not included or fell outside the pre-established range. A scientific committee ultimately decided whether the values could be accepted for inclusion in the registry.

Patients were required to produce at least one spontaneous sputum sample for microbiological analysis at each clinical visit, considering sputum as valid if <25 epithelial cells x field and >25 leukocytes per field were counted. Only patients who were able to give a minimum of two sputum samples each year were included in the study. The samples were homogenized (Gram-stained and plated on blood, chocolate, McConkey, and Saboreaud agar for culture). The cultures were expressed as CFU/mL (>10^3^ CFU was considered pathological, the same cut-off point used in other RIBRON studies).

The presence of an exacerbation was defined as a worsening of the typical bronchiectasis symptoms: cough, increased volume or purulence of sputum, dyspnea, hemoptysis, chest pain, or wheezing lasting more than 24 h that required treatment with antibiotics [27].

In terms of severity, exacerbations were classified into two groups: mild/moderate (if the patients required treatment with oral antibiotics) and severe (if hospital admission or intravenous antibiotic therapy was required, even if it was administered at home). Exacerbations were assessed in the year prior to inclusion in the registry and every year thereafter.

### 2.4. Statistical Analysis

Statistical analysis was performed using the SPSS program, version 18.0 (SPSS, Chicago, IL, USA). Quantitative variables were expressed as a mean (SD) and qualitative variables as absolute values and their corresponding percentages. The normality of the variables was established using the Kolmogorov–Smirnov test. For comparison of quantitative variables, Student’s t and Mann–Whitney U tests were used, according to the normality of variables. For comparison of qualitative variables, the chi-square test was used (as well as Fisher’s exact test when necessary). The comparison of three or more groups was carried out using an ANOVA or Kruskal–Wallis test, depending on the distribution of the variables. Variables of clinical interest and those with statistical significance (*p* < 0.1) in the univariate analysis were included as independent variables in a logistic regression model using the backward stepwise technique (Wald test).

Annual changes in the FEV1% value were assessed using a linear regression model with mixed effects for random intercepts. The model was adjusted using variables that were considered clinically significant: age, gender, initial FEV1 value, smoking status (ex-smoker, current smoker, or non-smoker), exacerbations occurring in the year before inclusion in the registry, hospitalizations during the same year, BMI, sputum color (mucous, mucopurulent, or purulent), CBI due to SA, aetiology, long-term antibiotic treatment (including inhaled antibiotics), macrolides, physiotherapy, use of inhaled corticosteroids, use of long-acting bronchodilators, presence of cystic BE, and number of affected lobes. When the correlation between two variables was more significant than 0.6 (Pearson) (collinearity), the analysis was performed by including these variables in two different models and evaluating the best model. Multivariate imputation techniques were used for missing data (less than 5% of data in any variable). A two-tailed *p*-value of less than 0.05 indicated statistical significance.

## 3. Results

### 3.1. General Characteristics

Four hundred and twenty-six patients from the RIBRON registry included between January 2015 and October 2020 met the inclusion criteria for the study. The baseline characteristics of the patients analyzed are shown in Table 1. The mean age was 66.9 years (16.2), and 297 (69.7%) patients were female, with a mean BMI of 25.1 (4.7) kg/m^2^ and an mean Charlson index of 1.74 (1.33). Regarding smoking habit, 290 patients (68%) were never smokers, 102 (24%) were ex-smokers, and 34 (8%) were current smokers. The median number of spirometries per patient was three (IQ 3–5). The mean value of FEV1 in the baseline spirometry was 2 L (0.76), the value of FEV1% being 78.8% (23.1). The mean value of FVC was 2.8 (0.91), 87% (20.9), and the mean value of FEV1/FVC was 71.2% (12.8). One hundred and seventy-two patients (40.4%) had airflow obstruction with FEV1/FVC < 0.7. The median number of sputum samples collected per year was three (IQ: 2–8), and the median follow-up was two years (IQ: 2–5), so all patients had a positive culture caused by SA during at least 24 months. The mean predictive FACED score was 1.62 (1.41), 2.62 (2.07) for the EFACED score, and 7.3 (4.5) for the BSI score. Seventy-seven patients (18%) presented with CBI due to SA, and 349 (82%) did not present any isolation of PPMs in sputum. The most frequent aetiology of bronchiectasis was post-infectious (37.1%), followed by COPD (9.6%), asthma (9.2%), systemic diseases (7.5%), and immunodeficiency (6.3%). In 69 patients (16.2%), the etiology was unknown. The number of exacerbations in the previous year was 1.44 (1.71), and the number of hospitalizations was 0.42 (1.11). Regarding lung function, all patients underwent a baseline spirometry test, followed by two subsequent annual monitoring tests. Sixty-nine patients (16.2%) underwent four spirometry monitoring tests.

### 3.2. Factors Associated with CBI by SA

#### Univariate Analysis

Table 1 shows a comparison between patients with CBI caused by SA in relation to those who were not colonized by pathogens during follow-up (univariate analysis). Patients with CBI caused by SA were younger (*p* < 0.0001) and had a poorer nutritional situation with a lower BMI (*p* = 0.024). In the previous year, it was observed that these patients more frequently presented with two or more exacerbations (*p* < 0.0001) and also presented more exacerbations (*p* = 0.019). The same results were observed in the first, second, and third years of follow-up (*p* = 0.020, *p* = 0.001, and *p* = 0.018, respectively).

As regards lung function, patients with CBI due to SA had lower levels of FEV1% at the time of inclusion in the registry (*p* = 0.021) and more frequently presented with bronchial obstruction (*p* = 0.042). According to analysis of the etiology of bronchiectasis, patients with genetic diseases (mainly primary ciliary dyskinesia) had CBI due to SA more frequently, but this CBI was significantly less frequent in other etiologies, such as bronchiectasis secondary to systemic diseases. Patients with CBI due to SA received treatment with macrolides and mucolytics more often, whereas the use of bronchodilators was unusual (*p* = 0.003).

### 3.3. Multivariate Analysis

Table 2 shows the multivariate analysis of factors associated with CBI due to SA. A lower age (OR: 0.97; 95% CI: 0.94–0.99; *p* < 0.001), lower FEV1 value% (OR: 0.98; 95% CI: 0.97–0.99; *p* = 0.035), higher number of affected lobes (OR: 1.53; 95% CI: 1.2–1.95; *p* < 0.001), and the presence of two or more exacerbations in the previous year (OR: 2.33; 95% CI: 1.15–4.69; *p* = 0.018) were observed as independent factors associated with CBI due to SA.

#### Impact of CBI Caused by SA on Lung Function

Table 3 shows lung function data (FVC, FVC%, FEV1, and FEV1%) of both patients with CBI due to SA and patients with no isolation of pathogens in respiratory secretions in the follow-up years. At the time of inclusion in the registry and the follow-up years, patients with CBI due to SA presented with a significantly lower FEV1% value when compared to patients with no isolation of PPMs.

Figure 2 shows a decrease in FEV1% in the entire sample included in the study (both those with and without CBI). The reduction in FEv1% was −0.31%/year (95% CI: −0.7; −0.07) (*p* = 0.110). When the reduction in FEv1% is analyzed separately in the group with CBI due to SA and in the group without pathogens (Figure 3), we observed that the reduction in FEV1% was −1.19% (95% CI: −2.09, −0.69) (*p* < 0.001) in the first group and −0.02% (95% CI: −0.07, −0.01) (*p* = 0.918) in the second group.

Performing a linear regression model (mixed effects) to determine which factors were associated with a more pronounced reduction in FEv1% in the overall group (including those with CBI due to SA and those with no PPM isolation), we found that the factors associated with a more accelerated reduction in lung function were age (*p* = 0.0019), use of inhaled corticosteroids (*p* = 0.004), presence of CBI due to SA (*p* = 0.007), female gender (*p* < 0.001), and the initial value of FEV1 (*p* < 0.001) (Table 4).

In Figure 3 shows data on the decrease in lung function adjusted to the baseline FEV1 value, comparing the group of patients with CBI due to SA (−1.19% (95% CI: −2.09, −0.69) *p* < 0.001) vs. the group with no isolation of pathogens (−0.02% (−0.07, −0.01) *p* = 0.918).

## 4. Discussion

According to the data obtained in our study, patients with CBI due to SA, compared to those who did not present isolations of PPMs in respiratory secretions, were younger, had lower FEV1% values the time of their inclusion in the registry, and a more significant extension of bronchiectasis, in addition to presenting more mild/moderate exacerbations in both the year before their inclusion in the registry and the years of follow-up. These findings may indicate a more significant state of persistent inflammation associated with the presence and persistence of SA in respiratory secretions.

However, we did not observe any increase in severe exacerbations and hospitalizations, which may have influenced the lack of differences in the severity scores for bronchiectasis between these two groups of patients. Once a harmful effect had been objectified in patients who did not present any isolation of PPMs in sputum, this impact could be assessed in subsequent studies to ascertain whether it was more or less significant than the impact caused by PPMs that had already demonstrated the same effect. In this respect, the association between the isolation of PPM and the presence of inflammation and chronicity in bronchiectasis has been described in patients with CBI caused by PA and *Haemophilus influenzae* (HI) [6,28].

The prevalence of SA isolation in the respiratory secretions of patients with non-CF bronchiectasis is variable, having been described in different series as a low prevalence of 4–7% [29,30,31,32,33,34]. In our series, 3.2% of the patients had CBI due to SA, and 7.8% had at least one SA isolation without meeting the criteria for CBI.

In relation to the etiology of bronchiectasis, it is important to bear in mind that in our study, infectious diseases were the most important cause of bronchiectasis, being described in 36.4% of cases with CBI due to SA, with tuberculosis being one of the most important infectious disease causing bronchiectasis. In this sense, recent studies emphasize the comorbidity of tuberculosis with different non-communicable diseases, such as bronchiectasis; in the recent past, host-directed therapies with drugs that modulate host responses have emerged as a novel and promising approach to treating tuberculosis, and it is likely that such treatments can avoid the development of complications [35].

It is important to consider the decrease in lung function throughout the three years of follow-up; when we analyzed the entire group of patients included in the study, we noticed a drop in FEV1% of 0.31%, whereas this decrease was much more pronounced in patients with CBI caused by SA, who showed a reduction in FEV1% of 1.19% (*p* < 0.001). Factors associated with a more accelerated reduction in lung function in the total group of patients included in the study were age (*p* < 0.001), use of inhaled corticosteroids (*p* = 0.004), CBI due to SA (*p* = 0.007), female gender (*p* < 0.001), and a low initial FEV1 value (*p* < 0.001). When we used another model in which the presence of two or more exacerbations in the previous year was included, without relating it to the CBI due to SA (because of the collinearity of both variables), we observed that the data were similar: the presence of two or more exacerbations in the previous year (*p* = 0.046) was linked to a quicker reduction in lung function.

These data were similar to those published by other authors, who achieved intermediate evolutionary data in patients with isolation of SA compared to patients with non-fermenting large, negative bacilli and those without isolations and concluded that isolation of SA is a marker of severity rather than a causal factor [34].

In a study carried out by Martínez-Garcia et al. [9] on the decrease in lung function associated with CBI due to BP in patients from the RIBRON registry, a reduction of 1.37% was observed, which is higher than that observed in patients with SA in sputum. This indicates an intermediate reduction of lung function values in patients with CBI due to PA or the absence of pathogens in respiratory secretions.

A review of the published literature reveals that some data have been described in CF patients, showing that the isolation of SA is widespread from the initial stages of the disease and has been related to a possible accelerated clinical deterioration [36,37,38,39,40,41]. However, the real impact of this infection in patients with non-CF bronchiectasis has not yet been determined [13,42], although some authors suggest that this pathogen could be associated with more severe disease [25,33].

Our data match those of Metersky et al. [34], who observed that patients presenting with CBI due to SA in sputum had lower FEv1% values at the time of inclusion in the registry and a higher number of exacerbations. However, these authors did not observe any association between the isolation of SA and the severity of the disease in the previous two years on the basis of a multivariate study. They considered that the presence of SA in respiratory secretions was a marker of severity rather than a causal factor. In patients in our study, the presence of CBI due to SA was associated with an increase in mild/moderate exacerbations in both the year before inclusion in the registry and the subsequent years of follow-up, and FEV1 values were also significantly lower compared to patients who did not present with isolations of pathogens in sputum. In our study, we only included methicillin-sensitive *Staphylococcus aureus*, and therefore, we cannot draw any conclusions about MRSA due to a lack of data.

When factors associated with a more significant reduction in lung function were analyzed in our study, we observed a more notable reduction in patients who presented with higher initial FEV1 values. This finding was previously observed in other studies analyzing the decrease in lung function in patients with CBI due to PA [9] and in patients with COPD [43]. It is therefore essential to consider the initial FEV1 value when analyzing the reduction in lung function. Other associated factors were age and female gender.

The association between a more accelerated reduction in patients receiving inhaled corticosteroids is worth mentioning. According to most therapeutic guidelines, the use of inhaled corticosteroids is discouraged in patients with bronchiectasis, apart from those with accompanying obstructive diseases [2,3]. In our study, patients who used inhaled corticosteroids were COPD patients or asthmatics, so it may be possible that the obstructive disease itself causes a more accelerated decrease in lung function. Another possibility is that patients who are more affected by the condition tend to use inhaled corticosteroids. In any case, following a multivariate model, the use of corticosteroids has been independently associated with a more significant loss of lung function, providing further data to discourage the use of inhaled corticosteroids in patients with bronchiectasis. As for CBI caused by SA, we constructed two models: one included CBI caused by SA, and the other considered the presence of two or more exacerbations in the previous year. Knowing that patients with CBI due to SA had a more significant number of exacerbations than patients with no pathogen isolations, it is more likely that the number of exacerbations may be the cause of the decline in lung function rather than the CBI itself, which would indicate that it could be a marker of severity rather than a causal factor. This suggests that exacerbations should be avoided in these patients in order to preserve their lung function.

### Limitations of the Current Study

As regards the limitations of our study, the sample of patients with CBI due to SA is small. However, this could also be a strength, as it means that patients included in the registry had to satisfy high-quality criteria and were well-characterized with spirometry monitoring tests conducted in a detailed and controlled manner. Another limitation is that we did not analyze the impact of methicillin-resistant *Staphylococcus aureus* (MRSA), which could have added some data comparing patients with CBI due to SA and patients with MRSA. Moreover, the follow-up took four years and was therefore not long-term. It is also important to bear in mind that some patients in the group with no pathogens in the sputum were not able to expectorate at all visits, which may add a selection bias, although all the patients included in the study gave at least two sputum samples each year. Finally, we excluded patients who died in the first year of inclusion in the registry, thus eliminating the most severe patients, with a consequent bias; nevertheless, we considered it appropriate to exclude these patients from the study, owing to the absence of monitoring data for them.

Although the study could have been approached from different perspectives, such as the comparison between the isolation of SA vs. non-SA or even CBI caused by SA vs. CBI caused by other PPMs, we nevertheless consider that the first step should be to assess whether CBI-SA causes any harm to the patient in comparison to the absence of PPMs, given that, although there is a considerable amount of information available for CF patients, the effect of SA on non-CF BE is unknown. It will then be important to assess the impact of CBI caused by SA on CBI caused by other PPMs, as well as to assess whether a single SA isolate may have an impact on patients with non-CF bronchiectasis.

## 5. Conclusions

In conclusion, patients with non-CF bronchiectasis with CBI due to SA are generally younger, with lower FEV1% values, a more significant extension of bronchiectasis, and a higher number of exacerbations of mild to moderate symptoms than those with no PPM isolation in respiratory secretions. The lower lung function values found upon inclusion in the registry indicate that these patients could previously have experienced a loss of lung function. This aspect was analyzed in our study, and we observed a more significant loss of lung function in patients who do not present any pathogens in sputum, indicating that the presence of two or more exacerbations in the previous year is associated with a more significant reduction in lung function. It is therefore essential to assess the need to treat CBI to avoid exacerbations and introduce management practices from the outset to prevent any loss of lung function in these patients. However, the small number of patients in our study does not allow us to draw significant conclusions; therefore, well-designed prospective studies are needed to enable us to draw firmer conclusions.

## Figures and Tables

**Figure 1 jcm-11-03960-f001:**
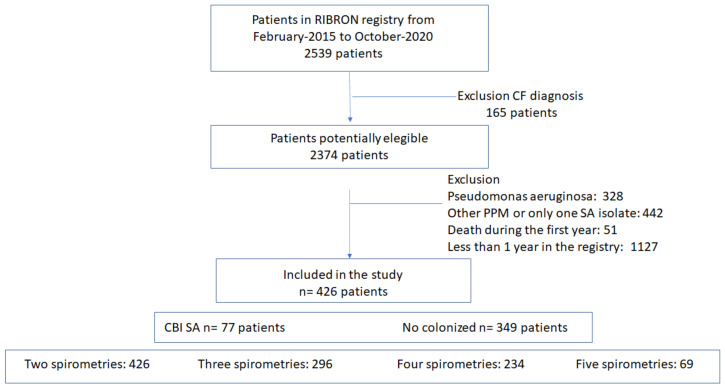
Flow chart of the patients included in the study. CF: cystic fibrosis; PPM: Potentially pathogenic microorganisms; CBI: chronic bronchial infection; SA: *Staphylococcus aureus*.

**Figure 2 jcm-11-03960-f002:**
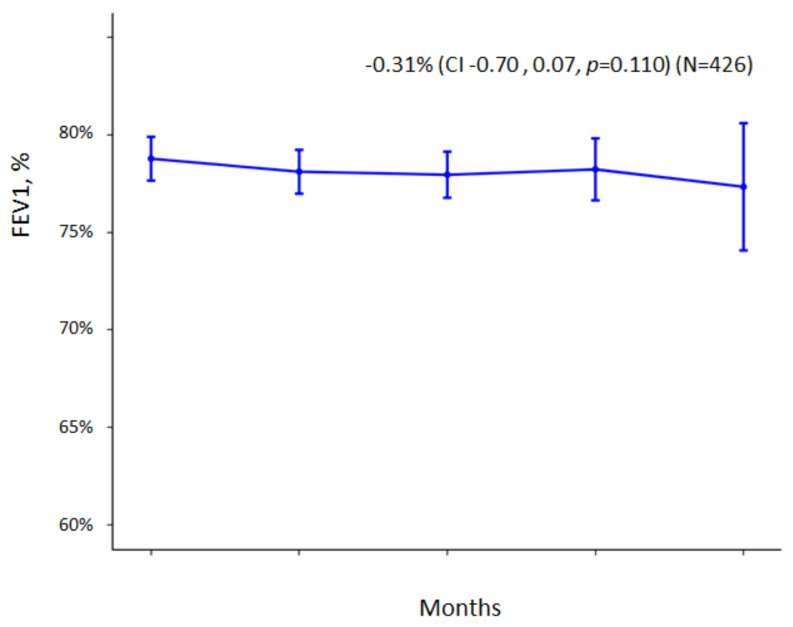
FEV1% decrease in the total group of patients included in the study (n = 426) (*p* = 0.110). FEV1: Forced expiratory volume one second; CI: Confidence interval.

**Figure 3 jcm-11-03960-f003:**
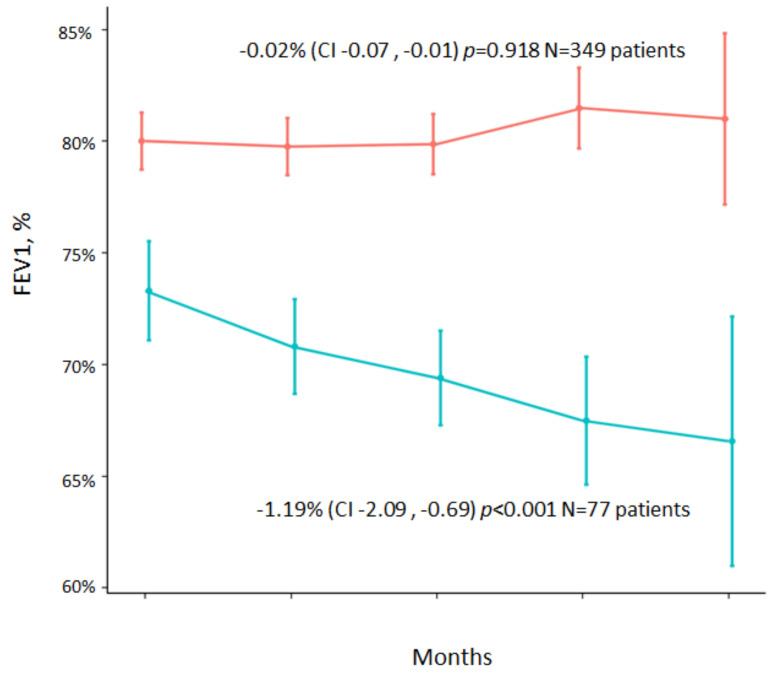
Decrease in FEV1% in the group of non-colonized patients (n = 349) (*p* = 0.918) and patients with CBI caused by SA (n = 77) (*p* < 0.001). FEV1: Forced expiratory volume one second; CI: confidence interval.

**Table 1 jcm-11-03960-t001:** General data of the sample and univariate analysis (comparison of patients with CBI due to SA and patients not colonized by pathogens).

	All Sample (N = 426)	CBI SA (N = 77)	Not Colonized (N = 349)	*p*
Age	66. 9 ± 16.2 (19–96)	60.2 ± 18.6	68.4 + 15.2	0.000
Age < 50 years	60 (14.1%)	19 (24.7%)	41 (11.7%)	0.003
Sex (male/female)	129 (30.3%)/297 (69.7%)	28 (36.4%)/49 (63.6%)	101 (29%)/248 (71%)	0.199
BMI (kg/m^2^)	25.1 ± 4.7 (14–44.1)	23.9 ± 4.96	25.3 + 4.56	0.024
Charlson index	1.74 ± 1.33 (0–12)	1.96 ± 1.97	1.70 ± 1.14	0.258
Exacerbations in previous year	1.44 ± 1.71 (0–12)	1.86 ± 1.99	1.35 ± 1.64	0.019
>2 exacerbations	159 (37.3%)	49.4%	34.5%	0.016
Hospitalizations in previous year	0.42 ± 1.11 (0–10)	0.26 ± 0.68	0.46 ± 1.18	0.051
>1 hospitalization	94 (22.1%)	16.9%	23.2%	0.226
Hemoptysis in previous year	0.49 ± 1.74 (0–21)	1 ± 2.9	0.37 ± 1.34	0.067
FACED score	1.63 ± 1.41 (0–6)	1.64 ± 1.31	1.63 ± 1.44	0.973
EFACED score	2.62 ± 2.07	1.97 ± 1.73	2.1 ± 1.8	0.594
BSI score	7.3 ± 4.5	7.1 ± 3.5	7.8 ± 3.9	0.630
Exacerbations (first year)	0.85 ± 1.20 (0–5)	1.30 ± 1.46	0.79 ± 1.01	0.020
Exacerbations (second year)	1.15 ± 1.28 (0–6)	1.60 ± 1.47	0.79 ± 1.08	0.001
Exacerbations (third year)	1.11 ± 1.56 (0–8)	1.80 ± 1.32	0.85 ± 0.88	0.018
Hospitalization (first year)	0.02 ± 0.14 (0–1)	0.12 ± 0.33	0.06 ± 0.37	0.320
Hospitalization (second year)	0.02 ± 0.14 (0–1)	0.07 ± 0.26	0.08 ± 0.42	0.902
Hospitalization (third year)	0.04 ± 0.19 (0–1)	0.19 ± 0.47	0.10 ± 0.39	0.272
Oxygen saturation	96.3 ± 2.3 (83–99)	96.5 ± 2.3	96.3 ± 2.3	0.529
Number of lobes	2.73 ± 1.45 (0–6)	3 ± 1.5	2.67 ± 1.45	0.063
FVC	2.8 ± 0.91 (0.56–5.92)	2.7 ± 1.5	2.74 ± 0.92	0.229
FVC%	87 ± 20.9 (26.7–161.2)	83.1 ± 17.4	85.8 ± 21.5	0.150
FEV1	2 ± 0.76 (0.41–4.98)	2.1 ± 0.74	1.98 ± 0.76	0.548
FEV1%	78.8 ± 23.1 (14.5–157.6)	73.3 ± 19.5	80 ± 23.7	0.021
FEV1/FVC	71.2 ± 12.8 (22.4–97.4)	68.5 ± 13.2	71.7 ± 12.7	0.047
Bronchial obstruction (FEV1/FVC < 70)	172 (40.4%)	50.6%	38.1%	0.042
Aetiology				0.000
Post-infectious	158 (37.1%)	28 (36.4%)	130 (37.3%)	
COPD	41 (9.6%)	5 (6.5%)	36 (10.3%)	
Asthma	39 (9.2%)	9 (11.7%)	30 (8.9%)	
Ciliary dyskinesia	16 (3.8%)	6 (7.8%)	10 (2.9%)	
Immunodeficiency	27 (6.3%)	6 (7.8%)	21 (6%)	
Systemic diseases	32 (7.5%)	1 (1.3%)	31 (8.9%)	
Inflammatory bowel disease	4 (0.9%)	3 (3.9%)	1 (0.3%)	
Others	40 (9.4%)	7 (9.1%)	33 (9.5%)	
Unknown	69 (16.2%)	12 (15.6%)	57 (16.3%)	
Treatment				0.003
Inhaled antibiotics	44 (10.3%)	9 (11.7%)	35 (10%)	
Macrolides	13 (3.1%)	7 (9.1%)	6 (1.7%)	
Inhaled corticoids.	76 (17.8%)	15 (19.5%)	61 (17.5%)	
Bronchodilators	136 (31.9%)	14 (18.2%)	122 (34.9%)	
Mucolytics	60 (14.1%)	17 (22.1%)	43 (12.3%)	

CBI: Chronic bronchial infection; SA: *Staphylococcus aureus*; BMI: Body mass index; BSI: Bronchial severity index; FVC: Forced vital capacity; FEV1: Forced expiratory volume one second; COPD: Chronic obstructive pulmonary disease.

**Table 2 jcm-11-03960-t002:** Multivariate analysis of factors associated with CBI due to SA.

	OR	CI 95%	*p*
Age	0.97	0.94–0.99	<0.001
Initial FEV1%	0.98	0.97–0.99	0.035
Number of lobes	1.53	1.2–1.95	<0.001
≥2 exacerbations in the previous year	2.33	1.15–4.69	0.018

OR: Odds ratio; CI: confidence interval; FEV1; Forced expiratory volume one second.

**Table 3 jcm-11-03960-t003:** Pulmonary function test data.

	CBI SA (N = 77)	Not Colonized (N = 349)	*p*
FVC (baseline)	2.97 ±0.91	2.75 ± 0.91	0.06
FVC (first year)	2.91 ± 1.1	2.72 ± 0.89	0.188
FVC (second year)	2.94 ± 0.92	2.69 ± 0.89	0.087
FVC (third year)	2.98 ± 0.95	2.75 ± 0.90	0.175
FVC% (baseline)	83.9 ± 17.6	87.8 ± 21.5	0.157
FVC% (first year)	80.8 ± 18.1	88.5 ± 21.1	0.015
FVC% (second year)	82.3 ± 17.5	88.8 ± 22.7	0.073
FVC% (third year)	80.6 ± 16.3	90.4 ± 20.4	0.012
FEV1 (baseline)	2.04 ± 0.74	1.98 ± 0.76	0.548
FEV1 (first year)	1.95 ± 0.71	1.93 ± 0.73	0.836
FEV1 (second year)	1.86 ± 0.67	1.91 ± 0.74	0.618
FEV1 (third year)	1.79 ± 0.69	1.96 ± 0.76	0.116
FEV1% (baseline)	73.3 ± 19.5	80 ± 23.7	0.021
FEV1% (first year)	70.8 ± 18.5	79.8 ± 23.6	0.002
FEV1% (second year)	69.4 ± 18.5	79.9 ± 24.9	0.001
FEV1% (third year)	67.5 ± 20.1	81.5 ± 23	0.000

CBI: Chronic bronchial infection; SA: *Staphylococcus aureus*; FVC: Forced volume capacity; FEV1: Forced expiratory volume one second.

**Table 4 jcm-11-03960-t004:** Factors associated with a more significant reduction in FEV1% in the total group of patients.

Variable	Coefficient β	95% CI	*p*
Age	−2	−3, −2	<0.001
Inhaled corticosteroids	−31	−52, −10	0.004
CBI caused by SA	−28	−48, −8	0.007
Gender	−42	−59, −24	<0.001
FEV1 baseline	−90	−93, −86	<0.001

CBI: Chronic broncuial infection; SA: *Staphylococcus aureus*; FEV1: Forced expiratory volume one second.

## Data Availability

All data are supported in RIBRON registry.

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
