# Peer review of "Impact of Chronic Bronchial Infection by Staphylococcus aureus on Bronchiectasis"

_jcm, 2022, doi:10.3390/jcm11143960_

Round 1
Reviewer 1 Report
With great interest I have read the paper of Clemente et al about the impact of chronic bronchial infection by SA on bronchiectasis. The paper is well written and there is a medical need for this topic.
I have some major and minor comments:
1. Abstract: the abstract is not adequate for a full paper.
2. Figure 1: Please add in the flow chart how many patients have SA.
3. A very important comparison is SA to PA and to other PPM. Why did the authors exclude these groups from analysis?
4. Ll124 please rewrite this sentence
5. Please explain chronic bronchial infection of SA in compare to colonization!
6. Does the patients had other SA infections?
7. Table 1 und table 2: I would advise to make one table from both.
8. Does patients with SA had the proof over 48 months?
9. The citation list (1-7) of the manuscript is overcrowded with self-citing. I recommend to change this. The readership of JCM is international.
The major problem is, that we still not know whether SA is a marker of severity rather tan a causal factor. To answer this question the authors have to compare with other PPM (for example H. influenza) and not only with a more healthy bronchiectasis patient group with no PPM.
Author Response
- Abstract: the abstract is not adequate for a full paper:
The Abstract has been changed.
- Figure 1: Please add in the flow chart how many patients have SA.
The Flow chart has been completed
- A very important comparison is SA to PA and to other PPM. Why did the authors exclude these groups from analysis?
Although the study could have been approached from different perspectives, such as the comparison between the isolation of SA vs. non-SA, or even CBI by SA vs. CBI by other MPP, we nevertheless consider that the first step would be to assess whether CBI-SA causes any harm to the patient in comparison to not having PPMs, given that, although there is a lot of information in CF patients, the effect of SA on non-CF BE is unknown. It will then be important to assess the impact of CBI by SA on CBI by other PPMs, and also to assess whether a single SA isolate may have an impact on patients with non-CF bronchiectasis.
This paragraph has been included in the discussion.
- Ll124 please rewrite this sentence
In the study, the presence of an exacerbation was defined as a worsening of the typical bronchiectasis symptoms: cough, increased volume or purulence os sputum, dyspnea, hemoptysis, chest pain or wheezing lasting more than 24 hours, which required treatment with antibiotics (27).
This paragraph has been included in the discussion.
- Please explain chronic bronchial infection of SA in compare to colonization!
We defined CBI as the presence of three or more consecutive positive cultures for the same PPM, always excluding an exacerbation period. We have not taken into account isolated cultures of SA that we have considered as colonization.
This explanation has been included in material and methods.
- Does the patients had other SA infections?
We have only included patients in a situation of clinical stability, without exacerbation, with chronic bronchial infection by SA, and therefore without other diseases related to this pathogen.
This explanation has been included in material and methods.
- Table 1 und table 2: I would advise to make one table from both.
Tables 1 and 2 have been unified.
- Does patients with SA had the proof over 48 months?
The median follow-up was two years (IQ: 2-5), so all patients have positive culture by SA during at least 24 months.
This explanation has been included in material and methods.
- The citation list (1-7) of the manuscript is overcrowded with self-citing. I recommend to change this. The readership of JCM is international.
Some references have been changed.
Reviewer 2 Report
1. What is the outcome of the study? Please include one or two lines in the abstract to make it complete.
2. Are there only females included in this study? Why? Please justify and provide the details.
3. Please improve the title of the figure1. Flow chart of what?
4. Bronchiectasis is a non-communicable disease, sometimes associated with other communicable lung infections such as tuberculosis. Please have a look at this article and discussion about this in the discussion. This is important.
5. https://pubmed.ncbi.nlm.nih.gov/30876870/
6. Did the authors check the studied patient’s history for other associated lung diseases?
7. Figure2. What about the controls here? Please include more details in the figure legend. What about the statistical significance. Please include the details in the figure legend too.
8. Figure3. What about the controls here? Please include more details in the figure legend. What about the statistical significance. Please include the details in the figure legend too.
9. The conclusions of the study are missing. Please include.
10. This study has a lot of limitations. Please include a separate section entitled “Limitation of the current study”.
Author Response
- What is the outcome of the study? Please include one or two lines in the abstract to make it complete.
The abstract has been completed.
- Are there only females included in this study? Why? Please justify and provide the details.
We include 297 (69.7%) females as you can see in table I.
- Please improve the title of the figure1. Flow chart of what?
The tittle has been changed: “Flow chart of the patients included in the study”.
- Bronchiectasis is a non-communicable disease, sometimes associated with other communicable lung infections such as tuberculosis. Please have a look at this article and discussion about this in the discussion. This is important.
https://pubmed.ncbi.nlm.nih.gov/30876870/
“In relation to the etiology of the bronchiectasis, it is important to be in mind that in our study infectious diseases were the most important cause of bronchiectasis, being described in 36.4% of cases with CBI due to SA, and being tuberculosis one of the most important infectious disease causing bronchiectasis. In this sense, recent studies emphasize the comorbidity of tuberculosis with different non-communicable diseases such as bronchiectasis and in the recent past, host-directed therapies with drugs that module host responses, have emerged as a novel and promising approach to treating tuberculosis and are likely that these treatments can avoid the development of complications like bronchiectasis (35)”.
I have included this paragraph in the discussion with the reference, but i dont know if this is exactly that the reviewer want.
- Did the authors check the studied patient’s history for other associated lung diseases?
Yes. In the RIBRON registry some variables are recorded (asthma, COPD, tuberculosis or other infectious disease). In table I associated lung diseases are registered.
- Figure2. What about the controls here? Please include more details in the figure legend. What about the statistical significance. Please include the details in the figure legend too.
In the figure 2 there are no controls. In this figure we have included the total group of patients. I have changed the figure with the total number of patients and the statistical significance. We include the details in the figure legend too.
- Figure3. What about the controls here? Please include more details in the figure legend. What about the statistical significance. Please include the details in the figure legend too.
Figure 3 includes two groups of patients, patients no colonized (n=349) and patients with CBI for SA (n=77). In this figure we can see the decrease in FEV1% in the two groups and the statistical significance. I have changed the figure.
- The conclusions of the study are missing. Please include.
The conclusions are included.
- This study has a lot of limitations. Please include a separate section entitled “Limitation of the current study”.
A section entitled “limitation of the current study” was included.
Round 2
Reviewer 2 Report
This study looks biased as there are only females included.
The authors should justify and discuss, why they did not include males in the present study?
Author Response
In the study we have include males and females.
In the results you can see that we have included 129 males and 297 females.
I have changed in table I the line of sex for a better understanding.
Best regards